# Identification of Candidate Genes for Rind Color and Bloom Formation in Watermelon Fruits Based on a Quantitative Trait Locus-Seq

**DOI:** 10.3390/plants11202739

**Published:** 2022-10-17

**Authors:** Siyoung Lee, Gaeun Jang, Yunseo Choi, Girim Park, Seoyeon Park, Gibeom Kwon, Byoungil Je, Younghoon Park

**Affiliations:** 1Department of Horticultural Bioscience, Pusan National University, Miryang 50463, Korea; 2Partner Seeds Co., Ltd., Gimje 54324, Korea

**Keywords:** bloom formation, fruit rind color, molecular marker, QTL-seq, watermelon

## Abstract

Watermelon fruit rind color (RC) and bloom formation (BF) affect product value and consumer preference. However, information on the candidate gene(s) for additional loci involved in dark green (DG) RC and the genetic control of BF and its major chemical components is lacking. Therefore, this study aimed to identify loci controlling RC and BF using QTL-seq of the F_2_ population derived by crossing ‘FD061129’ with light-green rind and bloom and ‘SIT55616RN’ with DG rind and bloomless. Phenotypic evaluation of the F_1_ and 219 F_2_ plants indicated the genetic control of two complementary dominant loci, *G*_1_ and *G*_2_, for DG and a dominant locus, *Bf*, for BF. QTL-seq identified a genomic region on Chr.6 for *G*_1_, Chr.8 for *G*_2_, and Chr.1 for *Bf*. *G*_1_ and *G*_2_ helped determine RC with possible environmental effects. Chlorophyll a-b binding protein gene-based CAPS (RC-m5) at *G*_1_ matched the highest with the RC phenotype. In the 1.4 cM *Bf* map interval, two additional gene-based CAPS markers were designed, and the CAPS for a nonsynonymous SNP in *Cla97C01G020050*, encoding a CSC1-like protein, cosegregated with the BF trait in 219 F_2_ plants. Bloom powder showed a high Ca^2+^ concentration (16,358 mg·kg^−1^), indicating that the CSC1-like protein gene is possibly responsible for BF. Our findings provide valuable information for marker-assisted selection for RC and BF and insights into the functional characterization of genes governing these watermelon-fruit-related traits.

## 1. Introduction

Watermelon is an economically important fruit belonging to the genus *Citrullus* Schrad. Ex Eckl and Zeyn of the Cucurbitaceae family. The cultivation area and production volume of watermelon (*C. lanatus* var. *lanatus* (Thumb)) accounted for 11,972 ha and 475,815 tons in Korea and 3,084,217 ha and 100,414,933 tons worldwide in 2019, respectively [1,2]. Watermelon (2n = 2x = 22) is native to South Africa and is generally classified into the cultivated-type species *Citrullus lanatus* var. *lanatus* (Thumb); ancient storage melon type *C. lanatus* var. *citroides* commonly grown in South Africa; perennial *C. colocynthis* growing in the sandy regions of North Africa, Southwest Asia, and the Mediterranean; perennial wild species *C*. *ecirrhosus* Cogn; and annual wild species *C*. *rehmii* De Winter [2,3]. The recent genome assembly of jubilee-type Chinese cultivar ‘97,103’ and crimson-type U.S. cultivar ‘Charleston Gray’ revealed the genome size of approximately 410 Mb and number of genes to be 22,571 [4,5].

Rind color (RC) is the main morphological trait for watermelon fruit, along with fruit shape, flesh color, and rind stripe pattern. Modern watermelon cultivars are characterized by various RCs, including gray, yellow, and different shades of green (light, medium, and dark) [6]. Inheritance of RC is difficult to explain due to the unclarified description in terminology for RC and inconsistency in gene expression depending on the genetic background used in different studies. Porter suggested the complete dominance of dark green (DG) rind over light green (LG) (described as yellowish white) and incomplete dominance over yellowish green [7]. Weetman reported that RC (LG vs. DG) and foreground stripe pattern were controlled by three alleles (dominant allele *D* for DG, recessive allele *d* for LG, and *ds* for rind stripe dominant to *d* and recessive to *D*) at a single locus *D* [8], which was renamed as G by Poole [9], or by two tightly linked different genes [6]. In contrast, Kumar and Wehner reported that two loci showing duplicate dominant epistasis were involved in formations of LG (gray) versus solid DG rind [10]. Recently, Li et al. identified the gene *ClCG08G017810* (*ClCGMenG*), encoding a 2-phytyl-1,4-beta-naphthoquinone methyltransferase protein, as a candidate gene that controls the complete dominance of DG color over the LG [11,12]. However, information on the candidate gene(s) for additional loci involved in DG RC is lacking.

Bloom is a white powder or outer wax layer of the epidermis of leaves and fruits, such as blueberries, plums, grapes, cucumbers, and pumpkins [13,14,15]. Bloom is mainly called fruit powder when it is found on the fruit surface, and it affects the appearance and product value of the fruit by preventing abiotic stresses such as drought and pathogen invasion into the fruit surface [13,16,17,18]. The bloom chemical composition differs depending on the crop species. The bloom of grapes mainly comprises aliphatic compounds, including alkanes and alkenes such as octadecane (C_18_H_38_) and eicosane (C_20_H_42_) [18]; meanwhile, silicon is the major component of cucumber and pumpkin bloom [19]. Various watermelon cultivars show variations in bloom or white powder formation on the fruit surface as well as the absence of bloom. However, the genetic control of bloom formation (BF) and its major chemical components is unknown.

Quantitative trait locus sequencing (QTL-seq) combines whole genome resequencing (WGRS) based on bulked segregant analysis (BSA) and next generation sequencing (NGS) to discover the QTL of a target trait [20]. In BSA, genotype differences are analyzed by pooling the DNA of progeny individuals showing extreme phenotypes for target traits in a segregation population obtained through crossbreeding. DNA bulk between different individuals in the target trait is effective in QTL research, because the genotype differs only at the locus related to the target trait [21]. WGRS can comprehensively reveal genetic variation between individuals by analyzing DNA sequences of the whole genome. Therefore, it is possible to analyze the exon region as well as the intron and untranslated regions and obtain a large number of nucleotide sequence variations [22,23]. QTL-seq does not require the development of molecular markers or the genotyping process for each individual in a segregated population for creating a genetic map, thereby remarkably reducing time and cost. QTL-seq has already been successfully applied to the study of agriculturally important traits of various crops [24,25,26], including the detection of candidate genes for semi-dwarfism and white-green flesh, and QTLs for *Fusarium* wilt resistance in watermelon [27,28,29].

In this study, we aimed to analyze the genetic inheritance of the RC (DG vs. LG) and BF (bloom vs. bloomless) traits of watermelon and identify the QTL and candidate genes for each trait based on QTL-seq. In addition, we developed molecular markers from the QTLs and candidate genes that can be used for marker-assisted selection (MAS) of these fruit-related traits. Phenotypic evaluation of the F_1_ and 219 F_2_ plants indicated the genetic control of two complementary dominant loci (*G*_1_ and *G*_2_) for DG and a dominant locus (*Bf*) for BF. QTL-seq identified a genomic region on Chr.6 for *G*_1_, Chr.8 for *G*_2_, and Chr.1 for *Bf*. The cleaved amplified polymorphic sequence (CAPS) for a nonsynonymous SNP in Cla97C01G020050 cosegregated with the BF trait in 219 F2 plants. The high Ca^2+^ concentration of bloom powder indicated that the CSC1-like protein gene is possibly responsible for BF. Our findings provide insights into the functional characterization of genes governing these watermelon-fruit-related traits.

## 2. Results

### 2.1. Mode of Genetic Inheritance

The RC of all five F_1_ plants was slightly paler DG (intermediate DG, IDG) than that of ‘SIT55616RN’, indicating an incomplete dominance of DG. In F_2_, nine LG and 210 DG, including IDG, colors were observed (Appendix A, Figure 1). The separation ratio of the phenotypes of DG (210 individuals) and LG (9 individuals) was 15:1 (χ^2^(0.05, 1) = 1.17, *p*-value = 0.19), indicating that the DG of ‘SIT55616RN’ is controlled by two complementary gene reactions.

The bloom was evident in ‘SIT55616RN’ and F_1_, showing complete dominance. In F_2_, 165 individuals showed bloom (B), whereas 53 were bloomless (BL) (Appendix A, Figure 1), which was consistent with the Mendel’s ratio of 3:1 (χ^2^(0.05, 1) = 0.07, *p*-value = 0.78), indicating that BF is regulated by a single dominant gene (Figure 1).

### 2.2. WGRS and Single Nucleotide Polymorphism (SNP) Detection

The NGS of DNA samples produced short reads covering a genome size more than 16 times larger than the reference genome sequence (97103 v.2, 425 Mb) in both parents and approximately 44 times larger in each F2-bulk (DG-, LG-, B-, and BL-bulk) (Appendix A). After preprocessing the sequences, filtering the reads, and comparing the read numbers between the bulks, the genome coverage for DG- and LG-bulk was approximately 34.5× and that for B- and BL-bulk was approximately 33.8× (Appendix A). The proportion of bases with a Phred score of 30 or higher (Q30) was 93% or higher for all DNA samples, indicating the reliability of base calling in NGS.

We obtained 156,919 SNPs from DG-bulk, 145,161 from LG-bulk, 157,055 from B-bulk, and 156,557 from Bl-bulk. Among them, 157,756 SNPs were comparable for RC and 159,692 for BF (Appendix A). Although the distribution of SNPs for each chromosome was slightly different between chromosomes, 6000–27,000 SNPs were obtained for each chromosome, thereby achieving a generally evenly distributed SNP.

### 2.3. QTL-Seq

The candidate QTL of the RC trait was detected at the ends of chromosomes 6 and 8 with 99% confidence interval (Figure 2a). The candidate QTL on chromosome (Chr.) 6 (RC-QTL-C6) was located at 0.270 Mb section (27,560,000–27,830,000 bp) and had 146 polymorphic SNPs and 22 genes. The candidate QTL region on Chr.8 (RC-QTL-C8) was detected in the 1.370 Mb section (26,830,000–28,200,000 bp), which harbored 441 polymorphic SNPs and 77 genes (Table 1 and Appendix A).

The SNP index of the QTL in Chr.6 was 0.21 in LG-bulk and 0.67 or higher in DG-bulk, and the delta SNP index value was −0.68 at the lowest, 0.84 at the highest, and above 0.47 on average. The SNP index of the QTL in Chr.8 was 0.15 in LG-bulk and 0.68 or higher in DG-bulk, and the average delta SNP index was over 0.54. In this QTL, 22 genes and 144 polymorphic SNPs were located, of which three SNPs were located in the coding region, but all were synonymous (Table 1). The genes carrying these three SNPs in the coding region were pentatricopeptide repeat-containing protein (Cla97C06G125670), expansin (Cla97C06G125700), and chlorophyll a-b binding protein (CAB protein, Cla97C06G125710). For CAB protein gene, three SNPs in a putative promoter region were additionally found. The candidate gene *ClCG08G017810* (Cla97C08G161570) of Chr.8 and its SNPs (C/G) reported by Li et al. were also found in the QTL region (27,994,152 bp) in this study [12] (Appendix A).

For BF, a single genomic region at the end of Chr.1 (BF-QTL-C1) was significantly associated with BF at a 99% confidence interval (Figure 2b). This genomic region was located in the 1.410 Mb section (32,100,000–33,510,000 bp). The average SNP indexes of BL-bulk and B-bulk were 0.96 and 0.38 or higher, respectively. The average delta SNP index was 0.58, with 674 polymorphic SNPs and 217 genes in this region (Table 1). Twenty-eight nonsynonymous SNPs were detected in the coding sequences, including genes for the trafficking protein particle complex subunit-like protein (Cla97C01G020540) and CSC1-like protein (Table 1 and Appendix A).

### 2.4. Cleaved Amplified Polymorphic Sequence Development and Genetic Mapping

#### 2.4.1. Rind Color

Nine CAPS markers were designed based on the SNPs in the genic regions, including four CAPS markers (RC-m3, RC-m4, RC-m5, and RC-m6) within RC-QTL-C6 and five (RC-m1 and RC-m2 from the upstream flanking region and RC-m7, RC-m8, and RC-m9 from the downstream flanking region) from the two flanking regions of RC-QTL-C6 (Table 2). For RC-QTL-C8, the CAPS marker RC-1D reported by Li et al. from the candidate gene (Cla97C08G161570) was directly used, and no other markers were designed [12].

A total of 219 F_2_ individuals were genotyped using all CAPS markers, and the results were compared with that of RC (Appendix A). Among the four CAPS developed within the RC-QTL-C6, Rc-m3, Rc-m4, and Rc-m5 (Figure 3a) cosegregated but recombined with the RC-m6 marker in two F_2_ plants. Higher recombination frequencies were observed between the three cosegregating markers and markers flanking RC-QTL-C6. Comparing marker genotypes with RC, the best fit was confirmed for the three cosegregating markers, including the chlorophyll a-b binding protein (CAB)-based marker, which was named locus *G*_1_ in this study.

Among 210 F_2_ plants with DG or IDG, 171 showed the marker genotype for ‘SIT55616RN’ (*G*_1_*G*_1_) or heterozygosity (*G*_1_*g*_1_), whereas the remaining 39 showed the marker genotype for ‘FD06112’ (*g*_1_*g*_1_). All nine F_2_ plants with LG RC showed marker genotype for ‘FD06112’ (*g*_1_*g*_1_). However, the RC-m6 marker genotype for the two recombinant F_2_ plants did not match the RC phenotype. In addition, the mismatch rate between the marker genotype and phenotype increased in the markers flanking RC-QTL-C6. The results implied that the gene for DG is the most tightly linked to the three cosegregating markers (RC-m3, RC-m4, and RC-m5) within RC-QTL-C6. Similarly, for the marker RC-1D in RC-QTL-C8, 163 F_2_ plants with DG or IDG showed the marker genotype for ‘SIT55616RN’ (*G*_2_*G*_2_) or heterozygosity (*G*_2_*g*_2_), whereas the remaining 47 showed the marker genotype for ‘FD06112’ (*g*_2_*g*_2_). All nine F_2_ plants with LG showed the marker genotype for ‘FD06112’ (*g*_2_*g*_2_). When the marker genotypes at *G*_1_ and *G*_2_ were combined, a high correlation existed between the intensity of the background green color (LG, IDG, DG) and the number of the dominant alleles. All F_2_ plants’ dominant homozygous for both loci (*G*_1_*G*_1_*G*_2_*G*_2_) showed the DG of ‘SIT55616RN’, whereas all F_2_ plants’ recessive homozygous (*g*_1_*g*_1_*g*_2_*g*_2_) showed LG of ‘FD061129’. Other F_2_ plants in the heterozygous state for at least one locus (*G*_1__*G*_2__) were DG of different intensity (DG or IDG), indicating that DG requires at least one dominant allele of the two loci (Table 3, Figure 3b).

#### 2.4.2. Bloom Formation

Eight CAPS markers were designed based on the SNPs in the genic region of BF-QTL-C1 (Table 2). All F_2_ plants were genotyped using these markers (Appendix A), and a genetic linkage map was constructed. As a result, a single locus for BF (*Bf*) was mapped between the markers BF-m3 and BF-m4, which showed a physical distance of 229,697 bp (Figure 4a). In this linkage group, *Bf* was linked to BF-m3 by 0.9 cM (two recombinant F_2_ plants) and BF-m4 by 0.5 cM (one recombinant F_2_ plant). In this flanking genomic region, 29 genes and 169 SNPs were observed within the gene (including the promoter). Among these SNPs, six SNPs were located in the exons of six genes (*Cla97C01G019850*, *Cla97C01G019900*, *Cla97C01G019940*, *Cla97C01G020050*, *Cla97C01G020060*, and *Cla97C01G020110*), of which three were nonsynonymous SNPs that alternate the genetic codons of amino acids (Figure 4a). Among these three SNPs, two CAPS markers, BF-m019900 and BF-m020050, were developed for SNPs located on genes *Cla97C01G019900* and *Cla97C01G020050* and used to genotype the 219 F_2_ plants (Appendix A). As a result, phenotype and genotype mismatch was found in one F_2_ plant with BF for BF-m019900 (Figure 4b), whereas the perfect phenotype–genotype match in all F_2_ plants was observed for BF-m020050, a marker for the gene *Cla97C01G020050* encoding a CSC1-like protein.

### 2.5. Determination of Bloom Powder Composition

The mean Ca^2+^ concentration in the bloom powder was 16,358 mg·kg^−1^, whereas that of Si concentration was 0.01 mg·kg^−1^, indicating that the major component of the bloom powder was Ca.

## 3. Discussion

The variation in fruit RC and BF in watermelon is important to consumers. However, selecting the intended trait can be challenging for breeders, because the expression of the trait is affected by the environment and the phenotype screen is only possible when the fruits are fully mature. The difficulties in selection based on phenotype can be circumvented by selection using molecular markers tightly linked to loci controlling fruit-related traits. In this study, we identified genomic segments harboring loci for DG RC and BF based on QTL-seq and developed CAPS makers that can be useful for MAS.

For RC, because the phenotype of F_1_ individuals was slightly lighter (IDG) than that of ‘SIT55616RN’ (DG), and the segregation ratio (15:1) of DG and IDG and LG was observed in the F_2_ population, we proposed an incomplete dominance inheritance for DG rind by two complementary loci (*G*_1_ and *G*_2_, named in this study). Our QTL-seq identified the two complementary loci at the terminal regions of Chr.6 and 8. These observations are similar to those reported by Kumar and Wehner, that the DG rind of ‘Mountain Hoosier’ and ‘Early Arizona’ was regulated by duplicate dominant epistasis of two loci (*G*_−1_ and *G*_−2_) [10]. However, the DG rind was completely dominant to LG, and no intermate DG rind was observed in F_2_ progeny.

Herein, we reported a novel locus (*G*_1_) on Chr.6 and its putative candidate gene *Cla97C06G125710* (CAB protein) responsible for the different shades of green RC in watermelon. The *Cla97C06G125710* gene encoding CAB was identified in the QTL region on Chr.6 (RC-QTL-C6) for the locus *G*_1_. Light-harvesting CAB protein (LHCP), an apoprotein found in major light-harvesting complexes of photosystem II (LHCII) in plants, is an important functional component in chloroplasts [30]. CABs are involved in light uptake, energy transmission to the reaction center of two photosystems (PS I and PS II), adjusting the distribution of the excitation energy between them, and maintaining the structure of the thylakoid membrane. CAB gene expression is affected by environmental conditions such as light intensity, temperature, and biotic or abiotic stresses [31,32]. Mutations in CAB are responsible for foliage albinism, reduced chlorophyll content, and aberrant chloroplast structures under high light intensity in tea (*C. sinensis*) [33,34]. Although little is known about the CAB genes in watermelon, our results and those of previous studies imply that mutations in CAB may be associated with green RC variations in watermelon. Further studies on the functional characteristics of the CAB gene and its possible interaction with *G*_2_ are necessary to understand the coloration of the watermelon rind.

The *ClCG08G017810* (*ClCGMenG*) gene on Chr.8 is the first candidate gene reported for DG RC [12]. In our study, this gene was located within the QTL region of Chr.8 (RC-QTL-C8) on the locus *G*_2_. The genotyping using *ClCG08G017810* gene-based marker RC-D1 [12] and WGRS data confirmed that the SNP in this gene reported by Li et al. existed between ‘SIT55616RN’ and ‘FD061129’, indicating that this gene corresponds to the locus *G*_2_ [12]. *ClCG08G017810* is an ortholog of the *MenG* gene in *Arabidopsis thaliana*, which encodes a 2-phytyl-1,4-beta-naphthoquinone methyltransferase protein involved in phylloquinone (vitamin K1) biosynthesis [35]. Phylloquinone is synthesized in plants, green algae, and some cyanobacteria and acts as a major electron transporter in photosystem I [36].

Bloom affects the drought resistance of sorghum [16] and acts as a barrier against pathogen invasion through the plant surface in several crops [13,14,15,16,17,18]. *Sobic.001G269200* (GDSL-like lipase/acylhydrolase) involved in the synthesis of fatty acids, a bloom component, is a candidate gene involved in the production and accumulation of bloom in sorghum [37]. In contrast, BF on cucumber is responsible for reducing its marketability by making the surface shine and skin color appear pale. The main component of cucumber bloom is Si [38,39]. Bloomless cucumbers are mainly produced using pumpkin graft stocks, which have limited Si absorption from the roots [38]. Many studies have been conducted on Si absorption and movement in rice [40,41,42]. In rice, the *Lsi1* gene located on Chr.2 is involved in Si absorption, whereas its movement (transport) from the root to the vascular tissue is controlled by *Lsi2* on Chr.3. The *Lsi6* on Chr.6, which is expressed in parenchymal cells of rice leaves, is involved in Si migration to the aboveground plant parts [43,44]. In pumpkin and cucumber, the *CmLsi1* and *CsLsi1* genes, homologs of the *Lsi1* in rice [40,41,42], are responsible for Si accumulation and BF [38,40]. However, little is known about the components of bloom and the genetic control of its formation in watermelon.

In our study, BF in ‘FD061129’ was conferred by a single dominant locus named *Bf*. QTL-seq and genetic mapping indicated the *Bf* is located within the 230 kb region at the end of Chr.1. Among the 28 genes in this region, no *Lsi* homologs were found, implying that Si is not a major bloom component in watermelon fruits. In contrast, three nonsynonymous SNPs in the gene encoding CSC1-like protein cosegregated with the *Bf*. The CSC1-like protein is an osmosensitive Ca-permeable cation channel in eukaryotes, which causes temporary changes in the number of Ca ions by osmotic stress [45]. In plants, CSC1-like protein is believed to be related to the signaling pathway underlying Marssonina blotch resistance in apples [46]. The expression level of CSC1-like protein increases under heat stress in maize [47]. In animals, CSC1 protein has been studied as a novel substructure of the Aurora B kinase complex, which regulates chromosome and cytoplasmic division in *C*. *elegans* [48]. When the CSC1-like protein gene in *Arabidopsis thaliana* (*AtCSC*) was expressed in the ovarian cells of Chinese hamsters, Ca ions increased, reaching a peak within a few seconds after hyperosmotic shock, and rapidly decreased within 1 min after treatment [45]. Ca-permeable activity activated by hypertonic stress has been reported in *Xenopus* oocytes [49], which was previously unknown in eukaryotes [50]. The powder analysis collected from the fruit surface of ‘FD061129’ revealed a remarkably high Ca^2+^ concentration (16,358 mg kg^−1^), whereas Si concentration was close to 0.0. The CSC1-like protein as a candidate gene for *Bf* and high Ca^2+^ concentration imply that Ca was the major bloom powder component in watermelon. Functional analysis of the CSC1-like protein gene through CRISPR/Cas9 will be a useful approach to elucidate the role of BF in watermelon fruits.

In conclusion, the DG RC of watermelon is controlled by two complementary loci (*G*_1_ and *G*_2_), which display incomplete dominance with possible environmental effects. The homozygous recessive genotype for both loci (*g*_1_*g*_1_*g*_2_*g*_2_) results in LG RC. QTL-seq identified *Cla97C06G125710* on Chr.6 and *ClCG08G017810* on Chr.8 as candidate genes for *G_1_* and *G_2_*, respectively. BF is controlled by a single dominant locus, *Bf*. QTL-seq and genetic linkage mapping identified the *Cla97C01G020050* gene encoding CSC1-like protein as a candidate gene for *Bf*, which may be involved in Ca accumulation as a white powder on the fruit surface. The CAPS markers developed based on these three genes will be useful for MAS of DG vs. LG RC and bloom vs. bloomless watermelon fruits. Functional analysis of these genes is necessary to elucidate their direct association with the phenotype.

## 4. Materials and Methods

### 4.1. Plant Material

For the genetic analysis of the RC and BF traits using QTL-seq, the inbred line ‘FD061129’ producing fruits with light-green RC and bloom and ‘SIT55616RN’ with dark-green RC and bloomless fruit were used as maternal and paternal parents, respectively. F_1_ progeny was created through artificial crossing between these lines, and F_2_ generation was created by self-pollination of five F_1_ individuals and combining the seeds produced from these plants (Figure 5).

### 4.2. Phenotyping and Genetic Inheritance Analysis

Phenotypic investigations of ‘FD06112’ and ‘SIT55616RN’ and their F_1_ and 219 F_2_ individuals were conducted in October 2020. After sowing in the nursery, all seedlings were grafted onto the gourd rootstock at half of the true leaf and planted in the soil at 20 cm intervals in a greenhouse located at Gimje, Korea. For phenotypic analysis, a single fruit from each individual was harvested 45 d after self-pollination, and RC and the presence or absence of bloom were visually examined. The RC was determined after wiping the fruit surface and classified into LG, DG, and IDG based on the parental lines and F_1_. BF on the rind surface was classified by the presence (B) or absence (BL) of a white powder.

To analyze the genetic inheritance of the two traits, a chi-square test was performed for the significance test (*p* < 0.05) of the Mendelian separation ratio in the F_2_ population using the Excel program (Microsoft, Inc., Redmond, WA, USA).

### 4.3. QTL-Seq

#### 4.3.1. Genomic DNA Extraction

Genomic DNA was extracted using two methods according to the purpose of experiments using young leaf samples stored at −80 °C after collection. For WGRS, leaf samples were frozen with liquid nitrogen and ground using a mortar, and then DNA was extracted using the GenEx™ Plant kit (Geneall, Seoul, Korea) following the manufacturer’s protocol. The quantity and quality of the extracted DNA was analyzed using 1% agarose gel electrophoresis (200 V, 45 min) and a Nanodrop1000 spectrometer (Thermo Fisher Scientific, Waltham, MA, USA). The final DNA concentration was diluted to 100 ng∙µL^−1^ for NGS.

The DNA samples used for CAPS genotyping was extracted using SDS. The collected fresh leaf samples were added to 600 µL SDS buffer, ground with Tissue Lyser II (Retsch, Haan, Germany), and placed in a water bath at 65 °C for 45 min. Further, 200 µL ammonium acetate was added, and the DNA was treated in an icebox for 20 min. After centrifugation at 13,000 rpm (13,475× *g*) at 4 °C, 600 µL of the supernatant was mixed with 600 µL isopropanol and 2.5 µL glycogen (10 mg·mL^−1^) and centrifuged under the same conditions. After discarding the supernatant, the pellets were washed with 200 µL 70% EtOH and centrifuged under the same conditions. The supernatant was removed, dried, and resuspended in 150 µL Tris-EDTA (TE) buffer. The concentration of the extracted DNA was measured using Nanodrop1000 and diluted to 10 ng∙µL^−1^.

#### 4.3.2. WGRS

WGRS was performed on each parental line and four F_2_-bulk DNA samples. For F_2_-bulk, DNA samples from F_2_ individuals showing phenotypes of the parental lines were bulked independently for the two traits. The DNA samples of nine plants of LG (LG-bulk), 40 of DG (DG-bulk), 40 of bloom (B-bulk), and 40 of bloomless (BL-bulk) were pooled in the same amount. Sequencing of these six DNA samples was performed using the Hiseq2000 and Nextseq NGS platform (Illumina, San Diego, CA, USA) using the paired-end sequencing (2×, 100–150 bp) method.

#### 4.3.3. QTL-Seq

High-quality short-read sequence data for QTL-seq were obtained uniformly from all DNA samples through sequence pre-processing using QTL-seq v.1.4.4 and FASTX-Toolkit v.0.0.13. The cleaned reads of ‘FD061129’ were aligned with the watermelon reference genome 97103 v.2 (CuGenDB, http://cucurbitgenomics.org) using the Burrows–Wheeler aligner (BWA; 0.6.1-r104) program, and the alignment product was converted into a sequence alignment and map (SAM)/binary alignment and map (BAM) file using the SAMtools v.0.1.16 program. Then, the reference sequence of ‘FD061129’ was prepared by replacing the nucleotide sequence of the SNP locus obtained through the SNP search and filtering process using the Coval v.1.4 programs. The cleaned reads of four bulk samples were aligned with the ‘FD061129’ reference sequence using the BWA (0.6.1-r104) program, and the alignment product was converted into a SAM/BAM file using the SAMtools program.

SNPs with a minimum read depth of 3 and a maximum of 75 were used to calculate the SNP index. The delta SNP index was calculated by comparing the SNP index values of the bulk sample at the same SNP position. SNPs with a read depth of 3 or more and an SNP index of 0.3 or more and 1 or less were used in both samples. A scatterplot of the entire chromosome was prepared by applying sliding window analysis (1 Mb window size and 10 kb increment) to the calculated delta SNP index value, and the QTL candidate region was searched at *p*-value < 0.01. SNPs and genes present in the QTL region were searched, and the syn/nonsynonymous status of each SNP was analyzed.

### 4.4. Genetic Linkage Mapping

#### 4.4.1. CAPS Marker Genotyping Analysis

The CAPS markers were designed to genotype the F_2_ population. The following SNP selection criteria were used for designing CAPS marker: (1) SNPs present in the QTL region and located on the gene and preferably within the exon region; (2) SNPs existing as evenly spaced as possible on the QTL region; and (3) possibly nonsynonymous SNPs. PCR primer for CAPS was designed using Primer3 v.0.4.0 and NEB cutter v.2.0.

#### 4.4.2. PCR and Electrophoresis

The PCR mixture (10 µL) was prepared using 10–20 ng genomic DNA, 1 µL 10× PCR buffer, 0.2 mM dNTP, 0.5 µL 10 pmol forward primer, 0.5 µL 10 pmol reverse primer, 0.5 U Taq polymerase (Solgent, Daejeon, Korea), and distilled water. PCR conditions were as follows: one cycle at 95 °C for 5 min, 35 cycles of denaturation at 94 °C for 30 s, annealing at TM of each primer for 30 s, extension at 72 °C for 1 min, and one cycle of final extension at 72 °C for 7 min. The restriction enzyme was added to the PCR amplicons according to the manufacturer’s instructions and then electrophoresed. Electrophoresis of the PCR product was performed using a 3% agarose gel (Philekorea, Seoul, Korea) containing 3 µL of ethidium bromide per 100 mL at 200 V for 1 h under UV light (Davinchi imaging system, Davinchi-K, Inc., Seoul, Korea).

#### 4.4.3. Genetic Linkage Map Construction

Genetic linkage maps for bloom traits were created at LOD 2.0, using the Kosambi map function in JoinMap v.4.1 (Kyazma, Wageningen, The Netherlands). Determining whether the bloom-trait phenotype was heterozygous in the F_2_ group in the matrix preparation of the program was not possible. Hence, ‘A’ indicates the genotype followed the maternal line or was heterozygous, ‘B’ indicates the genotype followed the paternal line, and ‘-’ indicates the genotype was missing.

### 4.5. Determination of Bloom Powder Composition

The bloom powder on the surface of the ‘SIT55616RN’ fruit rind was collected by gently scraping using a razor blade, washed with distilled water, and oven dried at 70 °C for 72 h. The bloom powder (2 g) was placed into a 100 mL glass flack and digested using 20 mL of ternary solution (HNO_3_/H_2_SO_4_/HClO_4_, 10:1:4, *v*/*v*) at 300 °C for 4 h for Ca and Si analysis. Ca and Si were quantified in four replicates using an atomic absorption spectrophotometer (AAS, AA-7000; Shimadzu, Tokyo, Japan), maintaining a very high accuracy, with detection limits of <0.002 and 0.001 mg·L^−1^ for Ca and Si, respectively.

## Figures and Tables

**Figure 1 plants-11-02739-f001:**
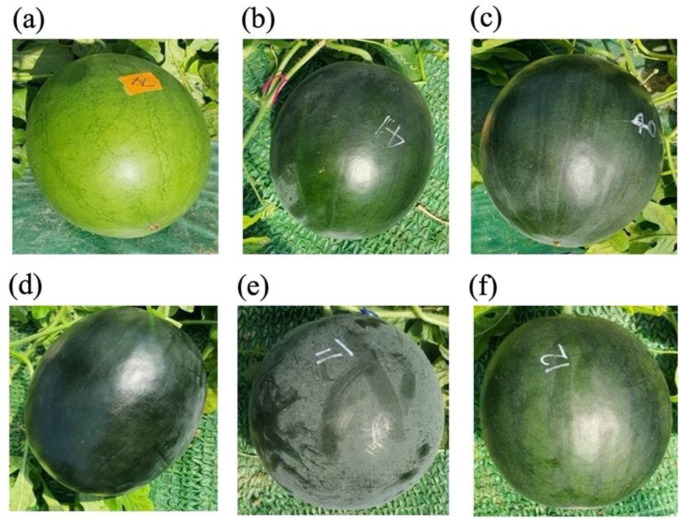
Watermelon fruit images of six F_2_ plants showing variations in the rind color (RC) and bloom formation (BF). (**a**) Light green rind (LG); (**b**,**c**) intermediate dark green rind (IDG); (**d**) dark green rind (DG); (**e**) bloom (B), and (**f**) bloomless (BL).

**Figure 2 plants-11-02739-f002:**
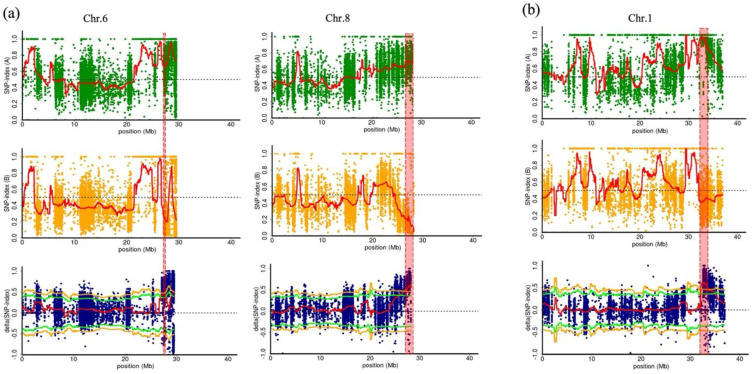
(**a**) Delta single nucleotide polymorphism (SNP)-index plots between dark green (DG)- and light green (LG)-bulk for rind color (RC) in chromosomes (Chr.) 6 and 8; (**b**) Delta SNP-index plots between bloom (B)- and bloomless (BL)-bulk for bloom formation (BF) in Chr.1. Each spot indicates an SNP. The green line indicates a probability value of <0.05, and the yellow line indicates a probability value of <0.01. The significant genomic regions (RC-QTL-C6 and RC-QTL-C8) are highlighted in red.

**Figure 3 plants-11-02739-f003:**
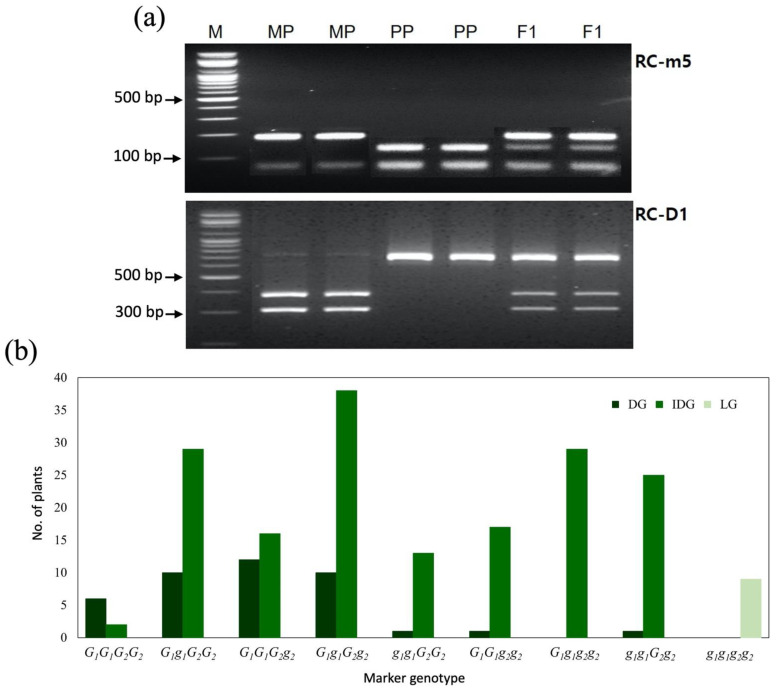
(**a**) Agarose gel image of CAPS marker RC-m5 at *G*_1_ (RC-QTL-C6) and RC-D1 at *G*_2_ (RC-QTL-C8) for the rind color of watermelon fruit. M, 100 bp size marker; MP, maternal parent ‘FD061129’; PP, paternal parent ‘SIT55616RN’; F_1_, progeny from the cross between MP and PP; (**b**) Distribution of F_2_ plants for rind color based on the classification of marker genotypes of RC-m5 and RC-D1. *G*_1_ and *g*_1_, allele for dark green and light green rind color, respectively, at the locus RC-QTL-C6; *G*_2_ and *g*_2_, allele for dark green and light green rind color, respectively, at the locus RC-QTL-C8.

**Figure 4 plants-11-02739-f004:**
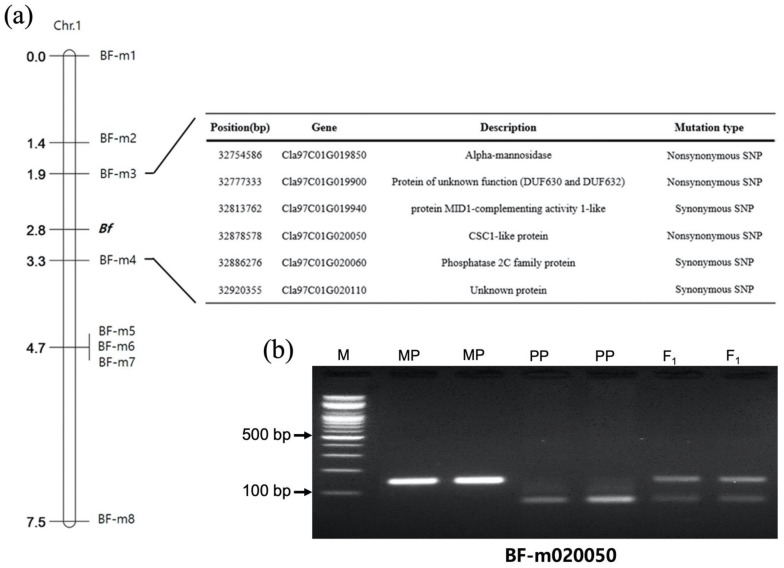
(**a**) A genetic linkage map showing the location of the locus (*Bf*) for bloom formation and gene annotation information and SNPs in coding sequences between the two cleaved amplified polymorphic sequence (CAPS) markers (BF-m3 and BF-m4) flanking *Bf*. (**b**) From the three nonsynonymous SNPs, two CAPS markers BF-m019900 and BF-m020050 were developed for SNPs located on genes *Cla97C01G019900* and *Cla97C01G020050* and used to genotype the 219 F_2_ plant SNPs. BF-m020050 cosegregated with *Bf* in this F_2_ population.

**Figure 5 plants-11-02739-f005:**
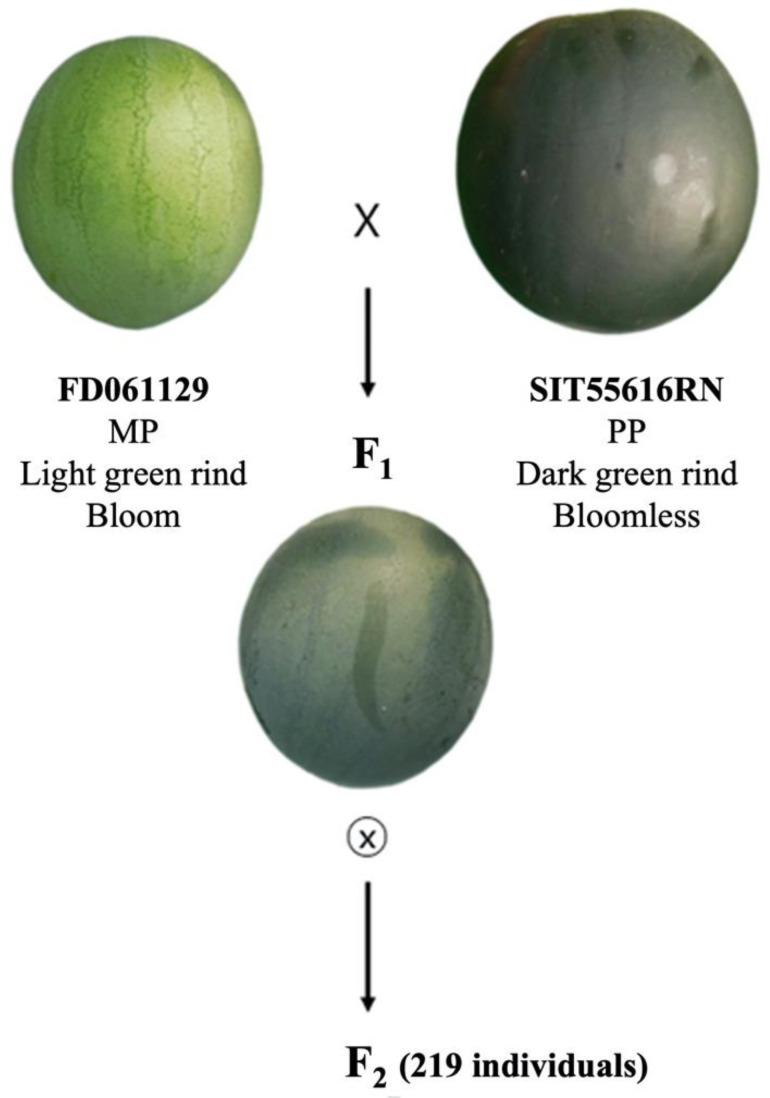
Watermelon-fruit images of maternal parent (MP) ‘FD061129’, paternal parent (PP) ‘SIT55616RN’, their F_1_ progeny, and the process used for the development of an F_2_ population.

**Table 1 plants-11-02739-t001:** List of quantitative trait loci (QTLs) for rind color (RC) (dark green vs. light green) and bloom formation (BF) (bloom vs. bloomless) in watermelon fruit identified by QTL-seq. The number of genes, single nucleotide polymorphisms (SNPs), and insertion/deletion (InDel) in the QTL regions.

Trait	QTL	Genomic Location	No. of Genes	No. of SNP		Nonsynonymous	Synonymous	No. of InDel	
				Total	Genic	SNP	SNP	Total	Genic
RC	RC-QTL-C6	Chr.6: 27.56–27.83	22	144	3	0	3	26	0
	RC-QTL-C8	Chr.8: 26.83–28.20	77	433	18	2	16	179	3
BF	BF-QTL-C1	Chr.1: 32.10–33.51	217	674	51	28	23	32	1

**Table 2 plants-11-02739-t002:** List of cleaved amplified polymorphic sequence (CAPS) markers developed from single nucleotide polymorphisms (SNPs) located on RC-QTL-C6 and BF-QTL-C1.

Marker	Gene ID	Gene Function	SNP Location	Forward Primer (5′→3′)	Restriction Enzyme	Expected Amplicon Size (PP/MP)^z^(bp)
RC-m1	Cla97C06G119680	dnaJ homolog subfamily C GRV2	20,950,898	F: TCCTCTCTGCTGCTTTGGTT	*Hpy*166II	186, 210, 35/396,35
				R: CATCCCACAAATGCCATGTA		
RC-m2	Cla97C06G123140	protein SMAX1-LIKE 4-like	25,477,052	GAGCCTATCGAGGCAATCAA	*Msp*I	157, 280/437
				CTTCCTTTCCGCTCAAACAC		
RC-m3	Cla97C06G125670	Pentatricopeptide repeat-containing protein	27,617,100	ACATGAATTACGCTGTTGTTGTTTT	*Mlu*CI	5, 80, 67, 37, 10, 22/5, 147, 37, 10, 22
				TAACTGCTTTCCCAACATAATTGAAC		
RC-m4	Cla97C06G125700	Expansin	27,636,308	ACTTCAATTTAGTTCTTGTAACCAACG	*Hpy*CH4V	116, 126, 47/242, 47
				AAAAACTCTGTTCCGTTTTGTTGTT		
RC-m5	Cla97C06G125710	Chlorophyll a-b binding protein, chloroplastic	27,640,473	GTTTTTGGGAGGCGAGTTATTAGTT	*Mlu*CI	47, 56, 118/47, 174
				AATGTGCTGCAATAGGTTGTCAAAT		
RC-m6	Cla97C06G125790	RWP-RK domain-containing protein	27,678,509	TGTAGGCTCATCAACTTCCTATGAG	*Bcc*I	35, 232/35, 116, 112
				GATGGTGGAGGAAATGAAGAAAATAGA		
RC-m7	Cla97C06G125810	promoter	27,691,376	TTGTTGATAGAGAGTGACATTTTGTT	*Mfe*I	226, 206/432
				CCCTTTAAACGCAGTAAACCA		
RC-m8	Cla97C06G125840	promoter	27,722,575	AACATGTTGTATTTCGTTGCATT	*Hpy*CH4V	19, 152, 254/425
				TTTGTGCCTATTTATGGTTGAA		
RC-m9	Cla97C06G127750	gamma carbonic anhydrase-like 2, mitochondrial	29,181,220	AAATGGCAGCTGTAGCTCGT	*Btg*I	272, 241/513
				AAAATTGCGAGTGCAGGAAT		
BF-m1	Cla97C01G019650	2OG-Fe(II) oxygenase family oxidoreductase	32,548,234	TTGTTTCCACCTGTTGTTTGTCTAA	*Bst*NI	253/72, 181
				TACTCATTCAACCGACAACAAAGAA		
BF-m2	Cla97C01G019840	26S proteasome non-ATPase regulatory subunit 4 homolog	32,731,607	ATTCTTCAATGGAGGAAATGGAGTC	*Hae*III	195, 104/299
				GACCTATCTAGAGAGCCCATGATTG		
BF-m3	Cla97C01G019840	26S proteasome non-ATPase regulatory subunit 4 homolog	32,741,354	TTGATTGTTATTCTGGGTTGTCAGT	*Rsa*I	177/103, 74
				ATGCTTTTAATTCCAGAAACTCACCG		
BF-m019900	Cla97C01G019900	Protein of unknown function (DUF630 and DUF632)	32,777,333	CTATGGGTTGCTGTTACTCGAGAT	*Hind*III	97, 91/188
				AATGTAGGTCTCTGCATTGGAAAAC		
BF-m020050	Cla97C01G020050	CSC1-like protein	32,878,578	TCTTTTGCTCTCCTCCAAATATACC	*Fok*I	13, 75, 59, 107/13, 135, 107
				TCTTTTGCTCTCCTCCAAATATACC		
BF-m4	Cla97C01G020120	Phospholipase D	32,925,042	AGCAAAATAAGAGCGAAGGAAAGAT	*Aci*I	74, 135/209
				GTCCATCACATTCGGTAATGGATAC		
BF-m5	Cla97C01G020220	Regulator of Vps4 activity in the MVB pathway protein	32,971,051	GTTGGTTAGGGTAGAATTAGCAGGA	*Hpy*188I	260/168, 92
				GAGCAAGCAGCAGTATTTTCATTTA		
BF-m6	Cla97C01G020540	Trafficking protein particle complex subunit-like protein	33,148,668	TGAGGTTAAAGTAAATCTGGGCAAC	*Rsa*I	88, 45, 42/133, 42
				ATCGAAGTTATTGCAGGAAATCAAG		
BF-m7	Cla97C01G020540	Trafficking protein particle complex subunit-like protein	33,155,492	ATCAGGGGTTCCTCTCAAAATTAAC	*Mlu*CI	18, 81, 45, 5, 26/18, 126, 5
				ATTGCAGTGGCAACTTAATCTGAAT		
BF-m8	Cla97C01G020630	promoter	33,210,953	TGCTCAATACATTTACAGCCTAATCA	*Mlu*CI	73, 27, 155, 41/73, 182, 41
				TGAGGGGTGAATCGATATTTACATTAT		

**Table 3 plants-11-02739-t003:** Phenotypic distribution of F_2_ plants based on the CAPS marker genotypes for two loci *G*_1_ and *G*_2_ controlling the rind color of watermelon fruits.

Genotype ^a^	Number of F_2_ Plants ^b^				Number of *G* Allele	Percentage of DG (%)
	DG	IDG	LG	Total		
*G* _1_ *G* _1_ *G* _2_ *G* _2_	6	2	0	8	4	75.0
*G* _1_ *g* _1_ *G* _2_ *G* _2_	10	29	0	39	3	25.6
*G* _1_ *G* _1_ *G* _2_ *g* _2_	12	16	0	28	3	42.9
*G* _1_ *g* _1_ *G* _2_ *g* _2_	10	38	0	48	2	20.8
*g* _1_ *g* _1_ *G* _2_ *G* _2_	1	13	0	14	2	7.1
*G* _1_ *G* _1_ *g* _2_ *g* _2_	1	17	0	18	2	5.6
*G* _1_ *g* _1_ *g* _2_ *g* _2_	0	29	0	29	1	0.0
*g* _1_ *g* _1_ *G* _2_ *g* _2_	1	25	0	26	1	3.8
*g* _1_ *g* _1_ *g* _2_ *g* _2_	0	0	9	9	0	0.0
Total	41	169	9	219		

^a^*G*_1_ and *g*_1_, alleles for dark green and light green rind color, respectively, at the locus RC-QTL-C6; *G*_2_ and *g*_2_, alleles for dark green and light green rind color, respectively, at the locus RC-QTL-C8. ^b^ DG, dark green; IDG, intermediate dark green; LG, light green.

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
