# Peer review of "Identification of Candidate Genes for Rind Color and Bloom Formation in Watermelon Fruits Based on a Quantitative Trait Locus-Seq"

_plants, 2022, doi:10.3390/plants11202739_

Round 1

Reviewer 1 Report

This paper identifies QTLs and associated markers for important consumer traits in watermelon. The methods and results are elaborate and clearly presented. I have few comments/clarifications for the authors.

1. Figure 5 shows 218 F2 individuals but the text mentions 219 F2 individuals.
2. Did all 219 F2 individuals come from selfing the same F1 plant or it is combined progeny from the five F1 plants? It is not clear in the text.
3. The authors could use Ca2+ when referring to 'Calcium concentration' or 'Ca ions' in the manuscript.
4. Inheritance studies normally include phenotypes (and chi-square tests) from a backcross population (to both parents) in addition to F2 phenotypes. Authors could mention why BC populations weren't necessary for their inheritance study?

Author Response

Response to Reviewers’ Comments

Dear Editor and reviewers:

On behalf of all the authors, I would like to thank you for taking the time and effort to review our manuscript and for your valuable comments. As suggested by the editor, we have thoroughly revised our manuscript based on your comments. Below, you will find our point-to-point responses to the comments.

Thank you again for your consideration.

Younghoon Park

Reviewer 1:

Comments and Suggestions for Authors

This paper identifies QTLs and associated markers for important consumer traits in watermelon. The methods and results are elaborate and clearly presented. I have few comments/clarifications for the authors.

  1. Figure 5 shows 218 F2 individuals but the text mentions 219 F2 individuals.

Answer: The total number of F2 plants used in this study was 219. The number in Fig.5 was changed to 219

  1. Did all 219 F2 individuals come from selfing the same F1 plant or it is combined progeny from the five F1 plants? It is not clear in the text.

Answer: The seeds from five F1 plants were combined to create an F2 population. The sentence was revised for clarification accordingly (line 378).

“F2 generation was created by self-pollination of five F1 individuals and combining the seeds produced from these plants (Figure 5).”

  1. The authors could use Ca2+ when referring to 'Calcium concentration' or 'Ca ions' in the manuscript.

Answer: ‘Ca concentration’ was changed to ‘Ca2+ concentration’ in the text. 

  1. Inheritance studies normally include phenotypes (and chi-square tests) from a backcross population (to both parents) in addition to F2 phenotypes. Authors could mention why BC populations weren't necessary for their inheritance study?

Answer: BC population could be useful for inheritance analysis. However, this population was not developed in this study and we only used the F2 population. Nevertheless, mendelian segregation ratios in F2 clearly showed the 2 complementary genes and single dominant gene effect for dark green rind and bloom formation trait, respectively.    

Reviewer 2 Report

1. Line 59-62, the author mentioned that "ClCG08G017810 (ClCGMenG) was regarded as the candidate gene controls the complete dominance of DG color over the LG". In this manuscript, the author also found a candidate region related with rind color. the same loci ?

2. The yellow rind color of watermelon was also reported, I did not see the information in the Introduction part.

3. Line 76-93, I think this paragraph is not related with this manuscript.

4. Line 424, Only nine plants of LG (LG-bulk), is it was enough for the QTL-seq?

5. For the rind color part, why the author did not use a linkage map to detect the chromosome region for RC trait?

Author Response

Response to Reviewers’ Comments

Dear Editor and reviewers:

On behalf of all the authors, I would like to thank you for taking the time and effort to review our manuscript and for your valuable comments. As suggested by the editor, we have thoroughly revised our manuscript based on your comments. Below, you will find our point-to-point responses to the comments.

Thank you again for your consideration.

Younghoon Park

Reviewer 2:

Comments and Suggestions for Authors

  1. Line 59-62, the author mentioned that "ClCG08G017810 (ClCGMenG) was regarded as the candidate gene controls the complete dominance of DG color over the LG". In this manuscript, the author also found a candidate region related with rind color. the same loci ?

Answer: In our study using QTL-seq approach, we could detect two loci for dark green rind color. A novel locus on Chr.6 was found in addition to the locus for ClCG08G017810 (ClCGMenG) gene on Chr.8, which was previously reported as a candidate gene for this trait.  

  1. The yellow rind color of watermelon was also reported, I did not see the information in the Introduction part.

Answer: A short explanation on the previous studies for yellow rind color inheritance was added in Introduction section. 

  1. Line 76-93, I think this paragraph is not related with this manuscript.

Answer: n this paragraph, authors intended to explain about the BSA-based QTL-seq approach for gene identification and some examples for its application to watermelon, because the QTL-seq was our main method in this study. 

  1. Line 424, Only nine plants of LG (LG-bulk), is it was enough for the QTL-seq?

Answer: This could be a small number. However, we could not add more sample into the DNA pool, because only nine F2 plants showed the light green rind color of their parental line (segregation ratio 15(dark green):1(light green). Nine was the maximum number we could bulked, but QTL-seq results reliable  

  1. For the rind color part, why the author did not use a linkage map to detect the chromosome region for RC trait?

Answer: According to our results, rind color is likely controlled by at least two loci. We agreed that QTL mapping and linkage analysis would reveal possible minor genes if they are involved in the expression of rind color variation. For QTL mapping, we thought that genome-wide markers such as SNPs would be required in addition to better measurement for the trait with detailed classification, which is generally possible through replicated field tests using RIL populations.  

Reviewer 3 Report

The paper entitled “Identification of Candidate Genes for Rind Color and Bloom Formation in Watermelon Fruits Based on a Quantitative Trait Locus-Seq” provided insightful information targeting the identification and mapping of genes responsible for certain important traits of watermelon using a unique sequencing method. The manuscript is well written. Authors should further improve and add recent and relevant citations to discuss their findings.  The linguistic aspect should be well checked throughout the text. In addition to these general comments, below are the specific comments about the changes necessary to the text.

Introduction

The objectives of the study should be highlighted in this section.

Conclusion

What is the main message behind this study that could be useful for agriculture?

Author Response

Response to Reviewers’ Comments

Dear Editor and reviewers:

On behalf of all the authors, I would like to thank you for taking the time and effort to review our manuscript and for your valuable comments. As suggested by the editor, we have thoroughly revised our manuscript based on your comments. Below, you will find our point-to-point responses to the comments.

Thank you again for your consideration.

Younghoon Park

Reviewer 3:

Comments and Suggestions for Authors

The paper entitled “Identification of Candidate Genes for Rind Color and Bloom Formation in Watermelon Fruits Based on a Quantitative Trait Locus-Seq” provided insightful information targeting the identification and mapping of genes responsible for certain important traits of watermelon using a unique sequencing method. The manuscript is well written. Authors should further improve and add recent and relevant citations to discuss their findings.  The linguistic aspect should be well checked throughout the text. In addition to these general comments, below are the specific comments about the changes necessary to the text.

Introduction

  1. The objectives of the study should be highlighted in this section.

Answer: The goal is this study was highlighted in the line 94-97 of page 2-3.

“In this study, we aimed to analyze the genetic inheritance of the RC (DG vs. LG) and BF (bloom vs. bloomless) traits of watermelon and identify the QTL and candidate genes for each trait based on QTL-seq”

Conclusion

  1. What is the main message behind this study that could be useful for agriculture?

Answer: The message was described in the last paragraph of the section ‘Discussion’, line 359-371, page 12

“In conclusion, the DG RC of watermelon is controlled by two complementary loci (G1 and G2), which display incomplete dominance with possible environmental effects. The homozygous recessive genotype for both loci (g1g1g2g2) results in LG RC. QTL-seq identified Cla97C06G125710 on Chr.6 and ClCG08G017810 on Chr.8 as candidate genes for G1 and G2, respectively. BF is controlled by a single dominant locus Bf. QTL-seq and genetic linkage mapping identified the Cla97C01G020050 gene encoding CSC1-like protein as a candidate gene for Bf, which may be involved in Ca accumulation as a white powder on the fruit surface. The CAPS markers developed based on these three genes will be useful for MAS of DG vs. LG RC and bloom vs. bloomless watermelon fruits. Functional analysis of these genes is necessary to elucidate their direct association with the phenotype.”

Round 2

Reviewer 2 Report

Thank you for the authors' efforts. Most of my concerns were explained.